# New Evidence for the Role of the Blood-Brain Barrier and Inflammation in Stress-Associated Depression: A Gene-Environment Analysis Covering 19,296 Genes in 109,360 Humans

**DOI:** 10.3390/ijms252011332

**Published:** 2024-10-21

**Authors:** Zsofia Gal, Dora Torok, Xenia Gonda, Nora Eszlari, Ian Muir Anderson, Bill Deakin, Peter Petschner, Gabriella Juhasz, Gyorgy Bagdy

**Affiliations:** 1Department of Pharmacodynamics, Faculty of Pharmaceutical Sciences, Semmelweis University, 1089 Budapest, Hungary; gal.zsofia2@semmelweis.hu (Z.G.); torok.dora@phd.semmelweis.hu (D.T.); eszlari.nora@semmelweis.hu (N.E.); petschner.peter@semmelweis.hu (P.P.); juhasz.gabriella@semmelweis.hu (G.J.); 2NAP3.0-SE Neuropsychopharmacology Research Group, Hungarian Brain Research Program, Semmelweis University, 1089 Budapest, Hungary; gonda.xenia@semmelweis.hu; 3Department of Psychiatry and Psychotherapy, Semmelweis University, 1083 Budapest, Hungary; 4Division of Neuroscience, School of Biological Sciences, Faculty of Biology, Medicine and Health, The University of Manchester and Manchester Academic Health Sciences Centre, Manchester M13 9NT, UK; ian.anderson@manchester.ac.uk (I.M.A.); bill.deakin@manchester.ac.uk (B.D.); 5Bioinformatics Center, Institute of Chemical Research, Kyoto University, Uji, Kyoto 611-0011, Japan; 6Research Unit for Realization of Sustainable Society, Kyoto University, Gokasho, Uji, Kyoto 611-0011, Japan

**Keywords:** blood-brain barrier, depressive symptoms, negative life events, *CSMD1*, *PTPRD*, biomarkers

## Abstract

Mounting evidence supports the key role of the disrupted integrity of the blood-brain barrier (BBB) in stress- and inflammation-associated depression. We assumed that variations in genes regulating the expression and coding proteins constructing and maintaining this barrier, along with those involved in inflammation, have a predisposing or protecting role in the development of depressive symptoms after experiencing severe stress. To prove this, genome-by-environment (GxE) interaction analyses were conducted on 6.26 M SNPS covering 19,296 genes on PHQ9 depression in interaction with adult traumatic events scores in the UK Biobank (n = 109,360) in a hypothesis-free setup. Among the 63 genes that were significant in stress-connected depression, 17 were associated with BBB, 23 with inflammatory processes, and 4 with neuroticism. Compared to all genes, the enrichment of significant BBB-associated hits was 3.82, and those of inflammation-associated hits were 1.59. Besides some sex differences, *CSMD1* and *PTPRD*, encoding proteins taking part in BBB integrity, were the most significant hits in both males and females. In conclusion, the identified risk genes and their encoded proteins could provide biomarkers or new drug targets to promote BBB integrity and thus prevent or decrease stress- and inflammation-associated depressive symptoms, and possibly infection, e.g., COVID-19-associated mental and neurological symptoms.

## 1. Introduction

Patients diagnosed with major depressive disorder (MDD) tend to exhibit heterogeneous phenotypic characteristics, which suggest different biological processes behind the etiology of this affective disorder [1]. The distinct biological routes result in the manifestation of various phenotypic or pathophysiologic subtypes of MDD. These subtypes could ensue from a combination of different personality traits, temperaments, and cognitive characteristics in interaction with various environmental influences and can be characterized by different clusters of symptoms, e.g., core symptoms (depressed mood, loss of interest), somatic symptoms (sleep, appetite, lack of energy, pain), and others (suicide, guilt) [2]. Additionally, subtypes of MDD can be classified based on the time of onset, sex, treatment outcome [3], symptoms [4], or comorbid disorders, too [5]. The effects of various stressors, mediated by distinct pathways, could also play a role in the emergence of depression [6]. Moreover, recent studies have implicated a potentially major role of immune system dysregulation in the development of MDD symptoms, denoting the possible influencing properties of inflammation on medical treatment as well [7]. The effects of chronic stress stimuli have also been associated with the presence of depressive symptoms and inflammatory processes [8]. However, the exact biological mechanisms behind this connection remain unclear. Considering the mechanism of action of currently used antidepressants, molecules of the monoamine systems (e.g., *MAOA*, *SLC6A2*, *COMT*, *SLC6A4*) are constant candidates for genetic studies [2]. However, the controversial outcomes of replication studies and the results from genetic meta-analyses on depression indicate that not all of these candidate genes would be substantial factors in MDD [2]. Taking into account these results and the heterogeneity of the disorder, in our study, we intended to reveal genes that contribute to the development of depression as stress-susceptibility factors.

An increasing number of studies focused on the blood-brain barrier (BBB) and the decrease in its integrity as potential key elements in depressive disorders, mediating the effects of psychological stress and peripheral inflammation [9]. Considering the crucial role of the BBB in the tight regulation of central nervous system homeostasis, likewise in pharmacological therapies, this structure could provide potential drug targets, aiming to restore its altered integrity preceding the development of symptoms or during MDD. Yet, targeting the BBB in human depression research can be challenging. Analyzing the human genome and single nucleotide polymorphisms (SNPs) is a prospective approach, though, as MDD shows a diverse scale of heritability, genetic components are more presumed to promote susceptibility for developing depressive symptoms instead of determining the evolution of the disorder [7]. Genetic polymorphisms in the promoter and enhancer regions may show their effects only during changes in gene expression induced by transient exposure to a variety of environmental factors, which—among others—could be stress effects. This is why the significance of genetic polymorphisms is much stronger in genome-by-environment (GxE) studies and in persons exposed to strong environmental factors [2,10]. Therefore, our aim was to investigate the potential contributions of genes regulating the rate of the expression or coding proteins that maintain, are part of the BBB, or are considered inflammatory-related in human stress-associated depression, providing a basis for more focused biomarker or drug target research in cases of stress- and inflammation-associated depression. For that, we conducted genome-by-environment interaction analyses (GWEIS) on current depressive symptoms score (measured by PHQ9 questionnaire) in interaction with adult traumatic life events in the genetic repository of UK Biobank (UKB). With the purpose of providing further evidence for the strong role of BBB and inflammatory factors in stress-induced depression; first, we arranged significant genes into three groups according to their previously published associations with the BBB, inflammation, or neuroticism (as this trait has been connected to stronger stress responses [11]) and compared the number of genes in each group; second, we calculated whether these numbers indicate enrichment of these genes in stress-associated depression, compared to all BBB- or inflammation-associated genes. These results were completed with replication in the independent NewMood cohort and sex-specific analyses.

## 2. Results

### 2.1. Significant Gene-Level GWEIS Results

Altogether, 109,360 participants were included in the genome-wide environment interaction analyses (GWEIS) on PHQ9 depression scores with adult traumatic events scores from the UKB cohort (Appendix A). Analyses were also performed in male and female subgroups separately, aiming to reveal potential sex differences. On the phenotypic level, significant correlations could be detected between PHQ9 depressive symptom mean values and adult traumatic event scores (Appendix A). At the gene level, 63 genes have been significantly (*p* < 2.591 × 10^−6^) associated with PHQ9 depression values in interaction with adult stress scores (Figure 1). The 63 genes, their encoded proteins, Z-scores, *p* values, and their previous associations to BBB, inflammation, or neuroticism are shown in Appendix A.

### 2.2. Enrichment of Blood-Brain Barrier (BBB)-Related Genes Among Significant Gene-Level GWEIS Results

As a comprehensive gene list of BBB is not available, we used recent gene expression data of the BBB [12] since significant changes in the expressions of genes have an impact on the cells themselves that take part in forming the BBB. In order to investigate the possible involvement of BBB-related genes in stress-associated depression, we applied MAGMA gene- and gene-set-level analyses on our SNP-level GWEIS results.

From the 63 significant genes, 17 could be connected to BBB structure and functions based on human midbrain gene expression data [12] (Figure 2a). The 17 significant BBB-related genes and the proteins they coded were: *CSMD1* (CUB and sushi domain-containing protein 1), *PTPRD* (Protein Tyrosine Phosphatase Receptor Type D), *RBMS3* (RNA Binding Motif Single Stranded Interacting Protein 3), *PRKG1* (Protein Kinase CGMP-Dependent 1), *DGKB* (Diacylglycerol Kinase Beta), *CHRM3* (Cholinergic Receptor Muscarinic 3), *PTPRG* (Receptor-type tyrosine-protein phosphatase gamma), *THSD7B* (Thrombospondin type-1 domain-containing protein 7B), *KCNJ6* (Potassium Inwardly Rectifying Channel Subfamily J Member 6), *PRR16* (Proline-Rich 16), *ADAMTS6* (ADAM Metallopeptidase With Thrombospondin Type 1 Motif 6), *ASIC2* (Acid Sensing Ion Channel Subunit 2), *GABBR2* (Gamma-Aminobutyric Acid Type B Receptor Subunit 2), *CLIC5* (Chloride intracellular channel protein 5), *DLGAP1* (Disks large-associated protein 1), *GPC5* (Glypican 5) and *ARHGAP18* (Rho GTPase Activating Protein 18).

This BBB-related gene set could be further extended with data from human postmortem microvascular samples from the temporal lobe [13]. Using this extended gene set, further significant BBB-related genes could be identified among our results (Appendix A).

In order to reveal the potential enrichment of BBB-related genes, the ratios of BBB-related genes among the significant results and all genes were compared. With Pearson’s χ^2^ (1, n = 19,296) = 38.04; *p* = 6.94 × 10^−10^, we could detect a 3.82 times increase in BBB-related genes among the significant GWEIS results. Analysis with BBB-related genes, based on human postmortem brain microvasculature expression [13], further supported the enrichment of BBB-related genes (results can be found in the Appendix A: “Chi-square statistics with BBB-related genes, based on human postmortem microvascular structures from the temporal lobe”).

### 2.3. Inflammatory-Related Genes Among the Significant Results

Beside the results from FUMA ‘GENE2FUNC’ gene set analyses—indicating the importance of the contribution of inflammatory-related gene sets and pathways (Appendix A)—we also patterned potential inflammation-related genes based on 162 inflammatory gene expression pathways [14] and inflammatory-related hallmark gene sets from the Molecular Signatures Database (MSigDB), which resulted in 23 significant gene-level hits (Figure 2b). From the 23 significant inflammatory genes, 5 genes (*PRKG1*, *DGKB*, *ADAMTS6*, *GABBR2,* and *GPC5*) were also part of BBB-related genes (Figure 3 and Appendix A). Table 1 shows the results and details of genes that were both associated with BBB- and inflammation.

The ratio between the observed number of BBB-related significant genes and their expected number, based on chi-square statistics (Pearson’s χ^2^ (1, n = 19,296) = 6.60; *p* = 0.0102), showed a 1.59 times enrichment of inflammatory-related genes among the significant GWEIS results.

### 2.4. Results of Sex-Specific Analyses

Because of the lower number of participants, and consequently the lesser power, in sex-specific analyses, the pattern of the significant genes was notably different among the whole sample, males, and females.

Four genes could be replicated in separate analyses in both male and female subsamples: *CSMD1*, *PTPRD*, *LSAMP,* and *NPAS3*. Among them, *CSMD1* and *PTPRD* could be considered BBB-related genes [12], although *LSAMP* and *NPAS3* were also found to be highly expressed in the brain microvasculature in transcriptomic analyses of human postmortem samples [13].

During sex-specific analyses, 44 genes (Appendix A) remained significant (*p* < 2.591 × 10^−6^) in male subjects (Appendix A) and 45 genes (Appendix A) in female subjects (Appendix A) after correction for multiple testing.

In male subjects, 29.5% of significant genes could be considered as BBB-related and 22.7% as inflammatory-related. In female subjects, 31.1% of significant genes could be connected to BBB functions and 24.4% to inflammatory gene sets. Based on that, in the case of male subjects, a 4.18 times increase in BBB-related genes could be observed among the significant ones, compared to the expected ratio of BBB-related genes included in the analysis [Pearson’s χ^2^ (1, n = 19,296) = 33.84; *p* = 5.99 × 10^−9^]. In female subjects, this increase in BBB-related genes among the significant ones was 4.40 times [Pearson’s χ^2^ (1, n = 19,296) = 39.60; *p* = 3.12 × 10^−10^]. Appendix A: “Chi-square test of inflammatory-related genes in men and women” contains the non-significant chi statistics for inflammatory-related genes in the case of men and women.

### 2.5. SNP-Level GWEIS Results

In the SNP-level analysis of the whole UKB cohort, 788 SNPs were significant (Appendix A), surviving correction for multiple testing (*p* < 5 × 10^−8^). As a result of the sex-specific analyses, 412 SNPs were significant (*p* < 5 × 10^−8^) in the case of male subjects (Appendix A) and 631 SNPs in female subjects (Appendix A).

### 2.6. Neuroticism-Related Genes Among the Significant Results

Four neuroticism-related genes could be detected among our significant results derived from the whole UKB cohort: *CDH13* [17], *LSAMP*, *CNTNAP2,* and *ERBB4* [18] (Appendix A). These genes were also significant in female subjects (Appendix A). In the male subgroup, 4 significant neuroticism-related genes (*RBFOX1* [17], *LSAMP*, *TENM2,* and *FHIT* [18]) (Appendix A) were found.

### 2.7. Sign Tests in the Independent NM Cohort Partially Replicate Significant Gene-Level Blood-Brain Barrier- and Inflammatory-Related Results

For replication purposes, we conducted GWEIS on 1753 participants of the NewMood (NM) Study, which included participants from Budapest, Hungary, and Manchester, United Kingdom (Appendix A) on the Brief Symptom Inventory (BSI) depressive symptoms value in interaction with last year’s stressors score.

Appendix A show the SNP- and gene-based GWEIS results on BSI depressive symptoms value in interaction with last year’s stressors score in the NM cohort.

Considering the huge difference in sample sizes (UKB n = 109,360; NM n = 1753) and power, we checked whether the gene-level GWEIS on BSI depressive symptoms values in interaction with last year’s stressors score had the same positive signs of Z statistics in the NM cohort compared to UKB GWEIS results, focusing on the 17 significant genes expressed in the BBB [12] and the 23 inflammatory-related genes of the GWEIS analysis in the UKB cohort.

As a result of the sign test of BBB-related significant genes, we found that 13 of the 17 genes had the same positive sign of Z statistics in the NM cohort, and among them, 4 genes (*CSMD1*, *ADAMTS6*, *DLGAP1*, *GPC5*) showed nominal significance as well (*p* < 0.05) (Appendix A). The sign test of inflammatory-related significant genes resulted in 15 identically positive Z statistics in the NM cohort, and among them, 4 genes showed nominal significance (*ADAMTS6*, *ADAMTSL1*, *TSPAN14*, *GPC5*) (Appendix A).

Appendix A: “Results of sign test in male and female subgroups separately”, contain the results of the sign test in male and female subsamples separately.

## 3. Discussion

In this large cohort study, using a hypothesis-free analysis (GWEIS), we identified 63 genes from the human genome (19,296 genes) that significantly interacted with stress and thus contributed to stress-associated depression. Several of these genes were new, which means they have never been connected to depression. Furthermore, our results provided important proof of the role of the blood-brain barrier (BBB) and inflammatory processes in stress-associated depression. Namely, we demonstrated that genes associated earlier either with BBB function or inflammation were more prevalent among our significant risk genes for stress-associated depression than could be expected by chance. Seventeen out of the 63 risk genes could be connected to the blood-brain barrier (BBB) and 23 to inflammatory processes, showing significant enrichment of BBB as well as inflammation-related genes (3.82 and 1.59 times, respectively (Figure 4)). In addition, four genes (CSMD1, PTPRD, LSAMP, and NPAS3) remained significant during the analyses in the whole UKB cohort and also in male and female subgroups, separately supporting the robustness of our findings.

The *CSMD1* gene encodes for complement inhibitor transmembrane protein (CUB and sushi-containing protein), which is expressed in ependymal [12] and endothelial cells of the brain microvasculature [13] and was found to be highly expressed in the hippocampus [19]. Its encoded protein has a role in inflammatory responses in the developing central nervous system in cell signaling [20] and in formulating the ratio between dopamine and serotonin metabolites in the cerebrospinal fluid [21]. Although none of the selected inflammatory-related gene sets contain *CSMD1*, based on literature data, it has a role in inflammatory responses, promoting the degradation of C3b and C4b complement factors [20]. As the elevated level of circulating complement factors may increase BBB permeability [22], the *CSMD1* gene—directly or indirectly—could also take part in this process. This gene has previously been associated with depression in a genome-wide association study (GWAS) [23], and animal studies with *CSMD1* knock-out mice [24] also supported its role in the development of anxiety- and depression-related symptoms. With our gene-level results, we supplemented these findings by revealing this gene’s stress susceptibility, which—based on a human maternal-fetal stress study [25]—might also be derived from epigenetic patterns (hypo-methylation) caused by high maternal cortisol levels.

The other BBB-related gene—based on human midbrain gene expression data [12], which remained significant in all GWEIS analyses—was *PTPRD*. This gene encodes for Protein Tyrosine Phosphatase Receptor Type D, a transmembrane protein, which could induce pre- and postsynaptic differentiation [26] and has suggested roles as a cell adhesion molecule [27]. *PTPRD* was shown to be expressed at a high level in ependymal- [12] and arteriolar smooth muscle cells of the BBB [13]. Although *PTPRD* was not included in the set of inflammatory-related genes derived from MSigDB, it has biological connections with the proinflammatory cytokine interleukine-6, which could indirectly promote the expression of *PTPRD* through the IL-6/JAK/STAT3 signaling pathway [28]. The genetic region of the *PTPRD* gene was previously associated with mood instability [29], different addiction phenotypes, restless leg syndrome, and obsessive-compulsive disorder [27].

Even though *LSAMP* and *NPAS3* genes were not considered BBB-related based on human midbrain gene expression data [12], they were found to be highly expressed in human postmortem brain microvasculature [13].

*LSAMP* encodes for Limbic System Associated Membrane Protein, which is part of the IgLON cell adhesion molecules superfamily, and it has a role in neuronal growth and axon targeting mediation with its cell membrane lipid anchor Ig domains [30]. It was found to be highly expressed in pericytes of the temporal lobe from human postmortem samples [13] and also in astrocytes of the BBB [31]. Its role in the maintenance of BBB’s integrity [32] is also supported by its ortholog in *D. melanogaster*, encoding Amalgam, which is critical to maintaining the BBB in the fly’s nervous structures [33]. Previous studies showed this gene’s stress susceptibility as, in rodents, perinatal stress caused alterations in the expression of *Lsamp*, which could be restored by an enriched environment [34]. This gene’s connection with depressive symptoms could also be detected in rodents since *Lsamp*-deficient mice showed altered neurotransmitter regulation with increased activity of the serotonergic system and imbalanced GABA-A receptor activity [35], and antidepressant treatment with fluoxetine elevated the level of *Lsamp* in mice, albeit, interestingly, not in rats [36]. Furthermore, in humans, polymorphisms of *LSAMP* were also associated with MDD, panic disorder [37,38], and neuroticism [18]. Additionally, a lower level of its encoded protein was associated with depressive symptoms in schizophrenia [39].

*NPAS3* encodes for Neuronal PAS Domain Protein 3, which is mostly a brain-expressed transcription factor, typically found in the cell nucleus [40]. It is involved in neurogenesis, cell proliferation, and also in the regulation of circadian rhythm [41]. This gene is expressed in fibroblasts of the BBB [12], and it was shown that *Npas3* deficiency impaired the differentiation of astrocytes and induced autistic-like behavior in mice [42]. A soluble form of Neuronal PAS Domain Protein 3 in serum appears to be associated with schizophrenia [40], and its gene showed pleiotropic effects among schizophrenia, MDD, and bipolar disorder [41].

We found 21 genes (from them, 7 BBB-related: *BARX2*, *COL6A2*, *PTPRM*, *SDK1*, *SLC38A5*, *TMEM132B*, *TRPC3*), which were only significant in women, and 39 genes (from them, 10 BBB-related: *BTNL9*, *CNTN4*, *CREB5*, *DPP10*, *EGLN3*, *EPHA7*, *RBFOX1*, *RHOBTB1*, *RUNX1*, *TENM2*), which were only significant in men, suggesting sex-specific mechanisms. One gene of note, which remained significant after correction for multiple testing in men but not in the case of the combined cohort or women, is *RUNX1*, which encodes for Runt-related transcription factor 1, expressed also in fibroblasts and smooth muscle cells of the BBB [12]. This transcription factor is a repressor of the claudin-5 tight junction protein of the BBB [43,44] and also has links with inflammatory mechanisms [14] as *RUNX1* is upregulated by TNF-α cytokines [45]. Significant gene-level GWEIS results of *RUNX1* in men on depressive symptoms in interaction with adult stressors could further strengthen our previous findings with functional polymorphisms of *CLDN5* (rs885985) and *IL6* (rs1800795) genes [46]. In our previous study, we found a significant association between the functional polymorphism of *CLDN5* and stress-associated depression only when taking into account the influencing properties of the widely studied polymorphism (rs1800795) of the interleukin-6 proinflammatory cytokine. However, the current results with *RUNX1* in the male subgroup of UKB could further support the important role of the *CLDN5* gene in stress-associated depression and BBB integrity, as this transcription factor repressing claudin-5 expression could be influenced by various inflammatory cytokines [45], suggesting that a biological pathway-level approach would be advisable during the investigation of the role of *CLDN5* in stress-associated depression. Stress susceptibility of this transcription factor has also been supported by studies revealing *Runx1* upregulation in stress-defeated mice [47] and discovering that *RUNX1* was a shared susceptibility gene between post-traumatic stress disorder (PTSD) and MDD [48].

We have been able to identify new potential genes that take part in the development of depressive symptoms, probably by making the individual more susceptible to, or resistant to, psychological stress stimuli. There were genes among our significant results that had been previously associated with the effects of stress. For example, increased *WDR70* (WD Repeat Domain 70) gene—which is expressed in endothelial cells of the BBB [12]—expression and its epigenetic modification were detected in social defeat-stressed mice [49] and *PTPRG* (also expressed in the BBB [12,13]) had previously been showed to be expressed by activated astrocytes during neuroinflammation [50]. Many theories exist as to how the effects of stress and the accompanying inflammatory processes disrupt the integrity of the BBB [51]. Besides the assumption of the major role of the appearance of peripheral inflammatory cytokines in the brain following psychological stress stimuli [52], it is also possible that microglial cells are activated by glucocorticoid and thus, stress-dependent mechanisms. This is supported by the circadian-dependent activation of these cells by glucocorticoids [53] that leads to the production and secretion of proinflammatory molecules, which could disrupt and modify the expression of tight junction, transporter, or any other proteins necessary for the integrity of this barrier [54]. Based on our results, the mechanism of neuroinflammation involving CD200 and its receptor [55,56] was also supported, as *CD200R1* (CD200 Receptor 1) remained significant after correction for multiple testing in the gene-level analysis of the whole UKB cohort.

Depression has already been associated either with elevated serum levels of proinflammatory cytokines or acute phase proteins, such as C-reactive protein (CRP) [57]; however, in MDD cases, the immune-related gene expression was independent of serum CRP levels [58]. Our results with inflammatory-related genes could provide a basis for biomarker research in depression.

The COVID-19 pandemic has also highlighted the connection between inflammatory mechanisms, the BBB, and various symptoms of cognitive dysfunctions, as besides neuroinvasive routes, the virus could also impair BBB integrity [59,60]. This trait of the infection, inducing immunovascular dysregulation and even structural changes in the brain [60], might cause the various symptoms of long COVID and affective disorders among them [61].

Although our GWEIS could not describe the interaction of the candidate genes and their causality in the disease mechanisms, the revealed BBB- and inflammatory-related genes might be promising targets in researching novel drugs with antidepressant effects. Based on randomized, controlled trials, immune targets (minocycline, celecoxib) were only effective in combination with antidepressants (superior to placebo plus antidepressants) [57], which indicated the importance of more nuanced inflammatory-related candidate selection.

Overall, the outcome of studies aiming to determine the genetic factors and variants involved in depression shows a lot of inconsistencies, and several biological, pathophysiological, and methodological shortcomings could be identified behind this phenomenon. The effects of individual genes and variants are weak; therefore, strict hypothesis-free approaches should be applied involving biological pathways and gene sets that can better capture the polygenic nature of MDD. Some inconsistencies could be explained by the inappropriate handling of the role of environmental stress in the models. In several case-control GWAS, genes encoding target proteins of currently used antidepressants remain non-significant, suggesting no main effect on depression and only an interaction with stress could be seen in a few studies [62]. The few significant genes in GWASs are related to neurogenesis, neuronal synapse, cell contact, and DNA transcription, and they are nonspecific for depression [2]. Some candidate genes in replicable GxE interactions are connected to the regulation of stress and the HPA axis [2]. Consequently, genes of traits and temperaments that were previously associated with stress sensitivity, such as neuroticism, had more straightforward interpretations in terms of depression [2]. Our current study tried to address some of these problems with a simple approach. First, in the GWEIS analysis, we use a hypothesis-free approach. Second, a very important environmental factor, stress, is included in the analysis. Third, instead of separate genes, gene sets for BBB, inflammation, and neuroticism were used and included in the enrichment analyses.

In conclusion, using genome-wide association analyses on depressive symptoms, taking stress as an influencing environmental factor, we could reveal novel supporting evidence for the involvement of BBB- and inflammatory-related genes and processes in stress-associated depression. As the four genes (*CSMD1*, *PTPRD*, *LSAMP,* and *NPAS3*), which remained significant also in the sex-specific analyses, have previously been implicated in other psychiatric disorders, pleiotropic effects could also be suggested. In summary, we have provided new candidates connected to the BBB and inflammatory processes for biomarker and antidepressant drug-target research in depressive disorders.

## 4. Materials and Methods

### 4.1. Populations/Participants

#### 4.1.1. UK Biobank (UKB)

In this study, we analyzed data from UK Biobank (application number 1602) and involved participants of white British ethnicity who had provided written informed consent, which had not been withdrawn until September 2023, and who passed all the phenotypic and genetic criteria mentioned below in sections “Phenotypes” and “Genotypes and imputation” (n = 109,360) (Figure 5). All procedures were carried out in accordance with the Declaration of Helsinki with ethical approval of the National Research Ethics Service Committee North West-Haydock (11/NW/0382, 21/NW/0157) and were previously published [63,64].

#### 4.1.2. NewMood (NM)

For replication purposes, combined Budapest and Manchester cohorts (n = 1753) were used from the NewMood Study (New Molecules in Mood Disorders, LHSM-CT-2004-503474, Sixth Framework Program of the European Union [65]). Recruitment and data handling were carried out in accordance with the Declaration of Helsinki and was approved by the Scientific and Research Ethics Committee of the Medical Research Council (Budapest, Hungary, [ad.225/KO/2005; ad.323-60/2005-1018EKU and ad.226/KO/2005; ad.323-61/2005-1018 EKU]) and the North Manchester Local Research Ethics Committee (Manchester, United Kingdom [REC reference no.: 05/Q1406/26]).

### 4.2. Phenotypes

Participants of UKB who filled out the “Depression in the last 2 weeks” questionnaire (Data Fields: 20507, 20508, 20510, 20511, 20513, 20514, 20517, 20518, 20519) based on the Patient Health Questionnaire depressive symptom scale (PHQ9) [66] and who answered questions about adult (age > 16) traumatic events (ATE) (Data Fields: 20521, 20522, 20523, 20524, 20525) in the “Mental Health Questionnaires” during the “Online follow-up” phase of the study were included in the analyses (n = 109,360). For each participant, the scores for each questionnaire (PHQ9 or ATE) were summed and divided by the number of answered questions.

In the NM cohort, current depressive symptoms were measured and self-reported based on the depressive subscale of the Brief Symptom Inventory [67] plus 4 items (“poor appetite”, “trouble falling asleep”, “thoughts of death or dying”, “feelings of guilt”). The sum of negative life events during the previous year was recorded based on The List of Threatening Experiences [68].

Age and sex, as basic sociodemographic information, were recorded (in UKB, Field IDs were: 21003 and 31, respectively) along with the questionnaires recording current mental health status in both cohorts.

### 4.3. Genotypes and Imputation

Genotyping of single nucleotide polymorphisms (SNPs) was performed from blood samples using two types of Axiom Arrays [69] in UKB and from saliva samples using Illumina’s CoreExom PsychChip in the case of the NM cohort. During genetic quality control (QC) steps [70], participants with high relatedness, extreme heterozygosity, or mismatching sex were excluded. Minor allele frequency (MAF > 0.01), Hardy-Weinberg equilibrium test (HWE > 0.00001), and missingness per marker and participant (missingness rate > 0.01 was excluded) filtering were applied to the whole genome. In addition, a priori to principal component analyses (PCAs), linkage disequilibrium pruning was performed (R^2^ of 0.2). Top 10 principal components (PCs) were calculated by Plink 2.0 based on the whole genome of participants having PHQ9 depression scores in the UKB, and having BSI depression scores in NM cohorts. In UKB, 6,258,585 SNPs, and in the NM cohort, 3,474,630 SNPs were included in the analysis.

### 4.4. Data Analysis and Replication

Plink 2.0 [71,72] was used for QC, PCAs, and genome-by-environment (GWEIS) linear regression interaction analyses (in additive models) in UKB and as a replication in the NM cohorts. Age, sex, and the first 10 PCs were included as covariates in the analyses (except in analyses conducted in male and female subjects separately, where sex was excluded). Due to the utilization of different chips, the genotyping array was also part of the covariates in UKB.

FUMA and MAGMA [73]—applied on GWEIS summary statistics—were used for gene-level analyses with the “UKB release2b 10k White British” version of the reference population, positioned using Ensembl v110 and with exact gene boundaries (gene windows = 0 kbp) as parameters. Significance thresholds were *p* = 5 × 10^−8^ (standard GWAS significance) in the SNP-level analyses. The significance threshold for gene-level results was determined with Bonferroni correction for multiple testing, which resulted in *p* = 2.591 × 10^−6^ (0.05/19,296) in gene-level analyses in UKB and *p* = 2.895 × 10^−6^ (0.05/17,274) in NM cohorts.

R version 4.3.0 [74] was used for descriptive statistics and for visualization purposes with packages: ‘tidyverse’ [75], ‘ggplot2’ [76], ‘wesanderson’ [77], ‘CMplot’ [78], and ‘jtools’ [79]. Chi-square tests were applied in order to reveal the potential alterations in the ratio of BBB and inflammatory-related genes among significant and a priori expected gene-level results.

During replication, GWEIS was conducted in NM on BSI depressive symptoms in interaction with negative life events during the previous year. Taking into account the smaller statistical power of the NM sample (using weighted means of β = 0.065 during the SNP-level power analyses in R; power at 10^−8^ significance level was 0.02% and power at 0.05 significance level was 48.06%), because of the huge difference in sample sizes, we did not expect GWEIS-level significant results. The replication was defined as the same direction (positive or negative) statistical coefficient sign of Z statistics as in the UKB GWEIS results.

### 4.5. Genes Connected to Blood-Brain Barrier Function and Maintenance

Recent publication with blood-brain barrier single-nucleus RNA expression data from 14 human midbrain samples (highly expressed genes per BBB cell type in the control group of the study) [12] was used as a reference for identifying BBB-related genes (n = 1364 from 7 cell types) in our analyses. As supplementary replication, considering the region-specific differences of the BBB [80,81], previously published data on post-mortem temporal lobe samples of control subjects with single-cell expression results of the human microvasculature was used (gene number = 3389 from 11 subclusters) [13].

### 4.6. Determining Pathways and Genes Connected to Inflammation Processes

FUMA and MAGMA [73] were used to explore potential pathways contributing to stress-associated depression, modeled by our GWEIS analyses. In addition, our significant gene-level hits were investigated further in order to reveal any potential involvement of inflammatory processes in stress-associated depression. For this analysis, genes from 162 inflammatory gene expression pathways [14] from C2 curated gene sets of MSigDB with additional inflammatory-related Hallmark gene sets of MSigDB were used (Appendix A).

### 4.7. Genes Connected to Neuroticism

Significant results from previous publications [17,18] were used as references in order to determine neuroticism-associated genes.

## Figures and Tables

**Figure 1 ijms-25-11332-f001:**
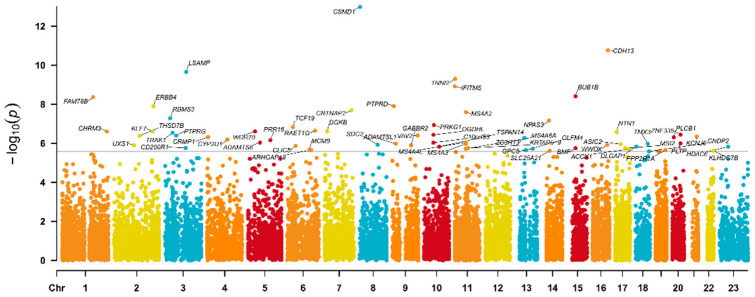
Manhattan plot of genes from GWEIS on PHQ9 depressive symptoms mean value in interaction with adult stressors in UKB. The x-axis represents the chromosomal location of genes across the genome (each chromosome is shown sequentially from chr1 to chr23). The y-axis represents the −log10 of the gene-based *p*-values, converted from the interaction between each genetic variant in a gene and adult traumatic events score on PHQ9 depression values in the UKB cohort. The horizontal black line indicates the gene-level genome-wide significance threshold (*p* = 2.591 × 10^−6^). Points above this line denote genes with significant interaction effects (n = 63). The most significant gene was *CSMD1* (Z stat = 7.3461; *p* = 1.02 × 10^−13^) on chromosome 8.

**Figure 2 ijms-25-11332-f002:**
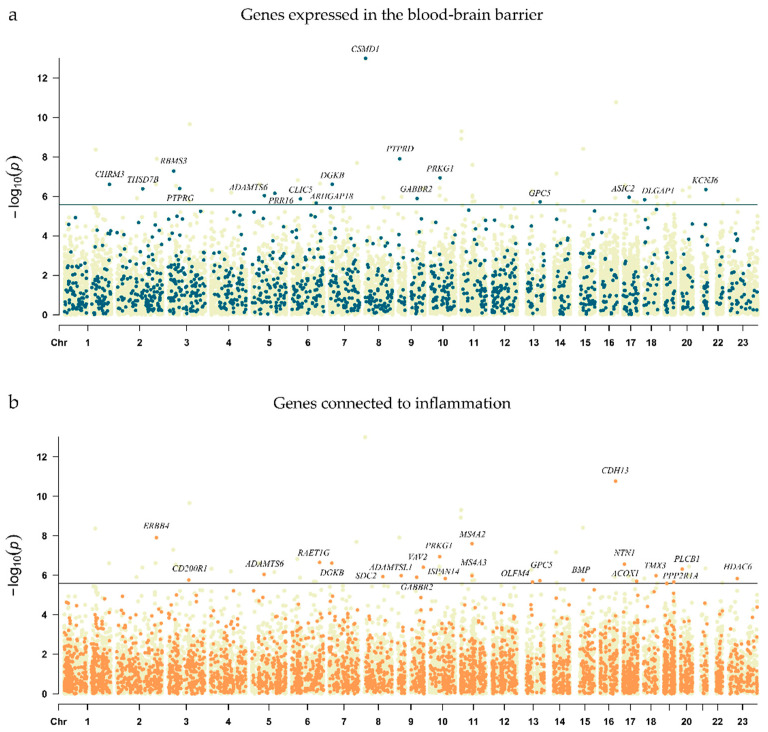
Manhattan plot of gene-based GWEIS on PHQ9 depression using adult traumatic events as interaction factor. The x-axis represents the chromosomal location of the analyzed genes across the genome (each chromosome is shown sequentially from chr1 to chr23). The y-axis represents the *p*-values (−log10) of the genes, resulting from the gene-based GWEIS on PHQ9 depression values in interaction with adult traumatic stress scores in the UKB cohort. The horizontal black line indicates the gene-level genome-wide significance threshold (*p* = 2.591 × 10^−6^). Names of significant genes are shown in the figures. (**a**) Selection of genes related to BBB was based on human gene expression data [12] and is highlighted in blue; (**b**) Selection of genes related to inflammatory processes was based on the Molecular Signatures Database (MSigDB) and is highlighted in orange.

**Figure 3 ijms-25-11332-f003:**
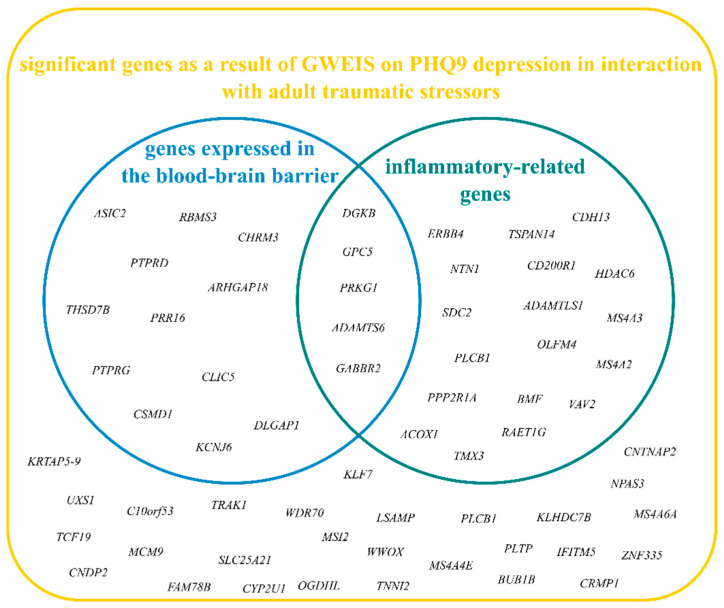
Venn diagram of significant gene-level results shows significant genes associated with BBB, inflammation, or both. Selection of genes expressed in the blood-brain barrier was based on Puvogel et al., 2022 [12]; selection of inflammatory-related genes was based on the Molecular Signatures Database (MSigDB).

**Figure 4 ijms-25-11332-f004:**
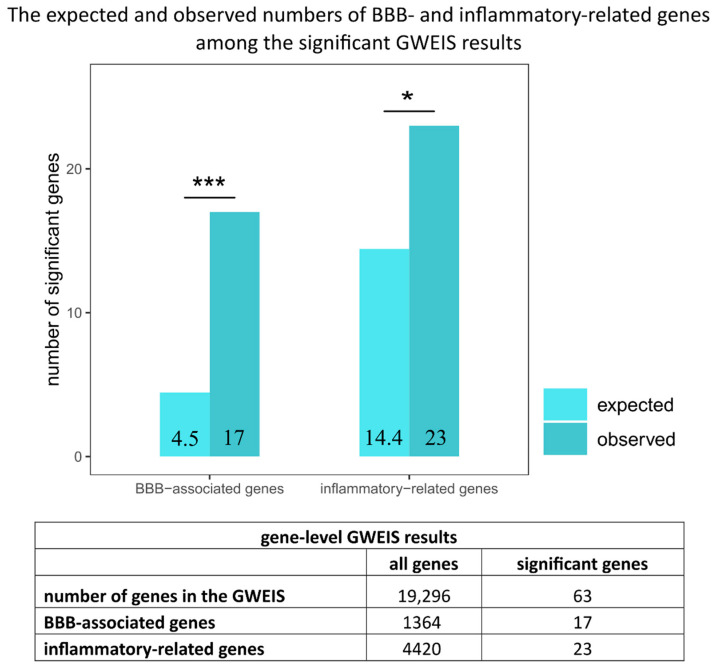
The expected and observed numbers of BBB- and inflammatory-related genes among the significant GWEIS results. Chi-square statistics were applied to the ratios of BBB-associated and inflammation-related genes among significant GWEIS results and all genes (see details in the Methods section). BBB-related genes were determined using gene expression data from human postmortem midbrain samples. Inflammatory-related genes were determined using MSigDB gene sets. Based on the expected and observed ratio of BBB-associated genes, a significant (***: *p* = 6.94 × 10^−10^) 3.82 times enrichment could be detected. In the case of inflammatory-related genes, a significant (*: *p* = 0.0102) 1.59 times enrichment was calculated.

**Figure 5 ijms-25-11332-f005:**
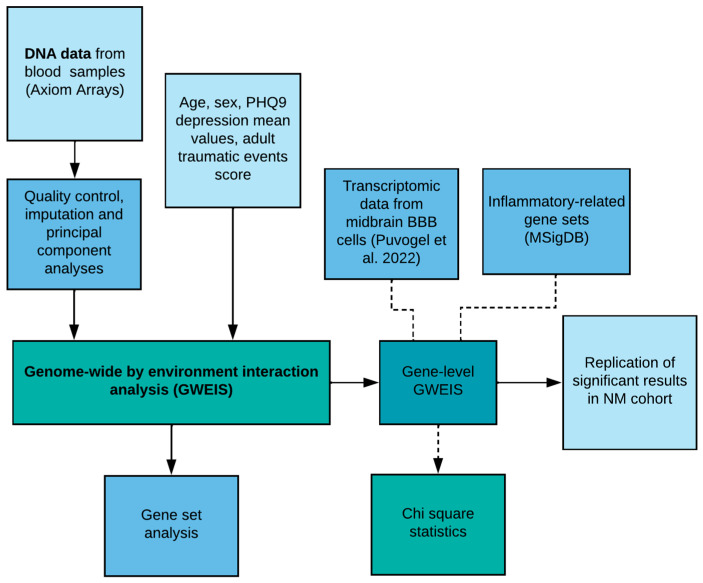
Workflow of experiments used in the study. In summary, the GWEIS was based on genetic- and phenotypic data of UK Biobank. SNP-gene- and gene set level analyses were conducted and the enrichments of BBB- [12] and inflammatory-related genes were calculated with Chi-square statistics using previously published data. Replication of significant gene-level results was performed in the NewMood cohort.

**Table 1 ijms-25-11332-t001:** Significant genes, which were considered as both BBB- and inflammatory-related, based on gene expression study [12] and MSigDB gene sets. Gene—symbol of the given gene; Chr—chromosome where the given gene is located; ZSTAT—Z-score converted from the gene-based *p*-value; P—*p*-value of the gene; Protein name—name of the encoded protein by the given gene; Functions—main functions of the encoded protein (* Based on Uniprot descriptions), including the relation of the gene to BBB-and inflammatory processes; Number of SNPs analyzed in the gene—the number of SNPs in the GWEIS, which could be mapped to the given gene; Top 5 most significant variants—the first 5 variants with descending *p*-values in that given gene and their annotations. The gene-level significance was determined based on the effect and significance of all of their SNPs. Note that the top 5 SNPs were not necessarily significant after applying the *p* = 5 × 10^−8^ standard GWAS SNP significance threshold.

List of Significant Genes Associated with Both BBB and Inflammation							
Gene	Chr	ZSTAT	P	Protein Name	Functions	Number of SNPs Analyzed in the Gene	Top 5 Most Significant Variants
*PRKG1*	10	5.1736	1.15 × 10^−7^	Protein Kinase CGMP-Dependent 1	“Serine/threonine protein kinase that acts as a key mediator of the nitric oxide (NO)/cGMP signaling pathway” *; expressed in ependymal cells, pericytes and SMCs [13] of BBB [12]; part of the adaptive immune system [15]	3425	rs56005445 (intron) rs111745376 (intron) rs78613136 (intron) rs78321473 (intron) rs78657836 (intron)
*DGKB*	7	5.0301	2.45 × 10^−7^	Diacylglycerol Kinase Beta	“Diacylglycerol kinase that converts diacylglycerol/DAG into phosphatidic acid/phosphatidate/PA and regulates the respective levels of these two bioactive lipids” *; expressed in BBB astrocytes and pericytes [12,13]; part of hemostasis gene set [15]	1839	rs12536670 (intron) rs1367776 (intron) rs1431534 (intron) rs1367778 (intron) rs12540165 (intron)
*ADAMTS6*	5	4.7719	9.12 × 10^−7^	ADAM Metallopeptidase With Thrombospondin Type 1 Motif 6	Plays a role in the formation, remodeling and homeostasis of the extracellular matrix (ECM): focal adhesions, epithelial cell interactions, microfibril assembly [16]; expressed in endothelial cells [13] and SMCs [12]; part of ECM regulators gene set [15]	594	rs76691813 (non coding transcript exon) rs6896524 (intron) rs55881998 (intron) rs11748932 (intron) rs62368972 (intron)
*GABBR2*	9	4.7032	1.28 × 10^−6^	Gamma-Aminobutyric Acid Type B Receptor Subunit 2	“Component of a heterodimeric G-protein coupled receptor for GABA” *; expressed in astrocytes [12]; part of genes, involved in G alpha signaling [15], inflammatory-related gene set [14]	1149	rs7869915 (intron) rs7853820 (intron) rs7853807 (intron) rs10739678 (intron) rs10760441 (intron)
*GPC5*	13	4.6224	1.90 × 10^−6^	Glypican 5	“Cell surface proteoglycan that bears heparan sulfate” *; expressed in astrocytes [12] and pericytes [13]; part of ECM regulators gene set [15]	3113	rs9516130 (intron) rs4771841 (intron) rs1572433 (intron) rs9301738 (intron) rs2065851 (intron)

## Data Availability

UK Biobank data are available for further research upon application to the data owners: UK Biobank (https://www.ukbiobank.ac.uk/, application number: 1602).

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
