# Peer review of "New Evidence for the Role of the Blood-Brain Barrier and Inflammation in Stress-Associated Depression: A Gene-Environment Analysis Covering 19,296 Genes in 109,360 Humans"

_ijms, 2024, doi:10.3390/ijms252011332_

Round 1

Reviewer 1 Report

Comments and Suggestions for Authors

This manuscript has the following shortcomings:

1. In a large cohort, we found that 63 genes interacted significantly with stress during adulthood in the outcome of depression. Seventeen of these could be connected to blood brain barrier (BBB) and 23 to inflammatory processes. Based on the number of all BBB associated genes and all genes included in our analysis, this meant a 3.82-times enrichment, providing evidence for the role of BBB functions in the development of stressinduced depression. Inflammatory-related genes among the significant results showed a 1.59-times increase, which supports previous assumptions about the role of inflammation in stress-induced depression. Four genes (CSMD1, PTPRD, LSAMP and NPAS3) remained significant during the analyses in the whole UKB cohort and also in male- and female subgroups separately.

The author should clearly state what the new findings of this article are? What is different from existing research? So as to facilitate readers' understanding.

2. Based on the number of all BBB associated genes and all genes included in our analysis, this means a 3.82-times enrichment, providing evidence for the role of BBB functions in the development of stress induced depression. Inflammation related genes among the significant results showed a 1.59-times increase, which supports previous assessments about the role of inflammation in stress induced depression. Please use a figure to represent this result for readers' convenience.

3. The figure legends for supplementary information are too simple, which is not conducive to readers' reading and understanding.

4. The arrangement of the official results and supplementary information results of this manuscript is unreasonable. The manuscript results 2.1 are all supplementary data charts. It is recommended to display the main text charts first, followed by supplementary data charts, or interspersed to facilitate readers' reading and understanding.

5. In the study (Supplementary information), results with p-values below 2.591 × 10⁻⁶ can be considered statistically significant, but should be interpreted with caution, as this threshold is more lenient compared to the standard p = 5 × 10⁻⁸ threshold. The results are best validated through replication studies or more stringent statistical methods to ensure their robustness.

Comments on the Quality of English Language

Minor editing of English language required.

Reviewer 2 Report

Comments and Suggestions for Authors

My suggestions:

1. The authors may mention a few genetic factors/variants, involved in MDDs.

2. The authors may explain briefly the subtypes of MDDs.

3. The authors may make a workflow of experiments in the Methods section.

4. The authors may add a table in the Results section on the significant genes, which are both expressed in BB and inflammatory genes, including their functions, and significant variants. 

5. Do the expression of the significant genes and the variants in these genes influence the clinical phenotypes of MDDs? 

6. Are there any plasma biomarkers that may be associated with inflammatory processes in MDDs? 

7. The authors may mention some drugs, which could target the inflammatory processes in MDD?

Round 2

Reviewer 1 Report

Comments and Suggestions for Authors

The manuscript also has the following issues:

1 Why are the reference numbers for the supplementary materials 12 and 13? But not 1, 2?

2 The table of contents in supplementary materials should include references.

Comments on the Quality of English Language

Minor editing of English language required.

Author Response

Dear Reviewer 1,

Thank you for pointing this out. We have changed the numbering of the references to 1 and 2 and included this chapter in the table of contents with page number in the supplementary materials document.

Reviewer 2 Report

Comments and Suggestions for Authors

The authors fulfilled my suggestions. Thank you.

Author Response

Dear Reviewer 2,

Thank you for your suggestions.